# Therapy of Locally Advanced and Oligometastatic Pancreatic Adenocarcinoma

**DOI:** 10.3390/cancers15245881

**Published:** 2023-12-18

**Authors:** Isabell Luisa Wahler, Alexander Damanakis, Nils Große Hokamp, Christiane Bruns, Thomas Schmidt

**Affiliations:** 1Department of General, Visceral, Cancer and Transplant Surgery, Faculty of Medicine, University Hospital of Cologne, University of Cologne, 50937 Cologne, Germany; isabell.wahler@uk-koeln.de (I.L.W.); alexander.damanakis@uk-koeln.de (A.D.);; 2Department of Diagnostic and Interventional Radiology, Faculty of Medicine, University Hospital of Cologne, 50923 Cologne, Germany

**Keywords:** oligometastasis, pancreatic adenocarcinoma, neoadjuvant chemotherapy

## Abstract

**Simple Summary:**

Pancreatic cancer remains a lethal disease despite a wide variety of therapeutic options. This review was initiated to provide an overview of existing therapies and future perspectives regarding the therapy of a patient collective that cannot undergo immediate curative treatment by surgery due to the extent of the disease at diagnosis.

**Abstract:**

Pancreatic adenocarcinoma is a lethal disease, and surgical resection remains the only curative treatment option. Unfortunately, upon primary diagnosis, only 15–20% of all patients with pancreatic ductal adenocarcinoma (PDAC) have localized disease that is eligible for operation. The remainder of patients either have borderline resectable or locally advanced disease or present with distant metastasis. In this review, we present a comprehensive overview regarding the current strategies and future directions in the multimodal therapy of locally advanced and oligometastasized pancreatic adenocarcinoma and discuss the benefit of surgery following neoadjuvant therapy in these patients.

## 1. Introduction

In 2020, pancreatic cancer accounted for almost as many deaths as cases (46,600 deaths vs. 49,600 cases) and was the seventh leading cause of cancer death worldwide [1]. It is projected that pancreatic cancer will be the third leading cause of cancer-related death in the US by 2030 [2]. 

More than 90% of cases are pancreatic ductal adenocarcinoma [PDAC] [3]. Despite progress in oncological therapies, the 5-year survival rate of pancreatic ductal adenocarcinoma remains at 10% [4]. One major challenge is that more than 50% of patients diagnosed with pancreatic adenocarcinoma have distant metastasis at the time of diagnosis, and only 15–20% of diagnosed patients are eligible for surgical resection, to date the only curative treatment option [5]. 

The clinical and pathological stagings of PDAC are primarily based on the TNM stage, according to UICC [6]. Local resectability defines a patient’s eligibility for surgery. According to NCCN, there are three categories defining local resectability: resectable disease, borderline resectable, and locally advanced. Those categories are characterized by the involvement of the hepatic arteries, celiac trunk, superior mesenteric artery, and the portal vein, as well as the superior mesenteric vein (see Table 1 for criteria). The gold standard for the evaluation of vascular involvement is a contrast-enhanced thin-section CT scan [7,8]. Locally advanced cases are considered not eligible for surgical resection because of extensive vascular involvement. Patients with distant metastasis (stage IV, cM+) should not undergo surgical resection, regardless of the resectability of the primary tumor, according to current international guidelines [9]. But a new dynamic has emerged over the last ten years, with recent advancements in multiagent chemotherapy and multimodal treatment concepts showing conversion to resectable disease in locally advanced cases. 

Retrospective data suggest that there might be a benefit even for selected stage IV patients after chemotherapy when performing surgical resection of the primary tumor and the metastases [10,11]. The latter group is widely recognized as having oligometastatic disease, even though a universally accepted definition still does not exist [12]. Also, progress in operation techniques has made vascular replacement and multivisceral resection a safe option in specialized high-volume centers for pancreatic surgery [13,14].

The main driver of this new dynamic has been considerable progress in the palliative and adjuvant treatment of patients with PDAC in the last decade. In 2011, Conroy et al. first reported improved overall survival in patients with metastatic pancreatic cancer treated with multiagent chemotherapy (FOLFIRINOX) when compared to standard-of-care gemcitabine monotherapy of 11.1 vs. 6.8 months. Progression-free survival was increased from 6.4 months to 3.3 months in the FOLFIRINOX group [15]. One limitation of this combination therapy is its higher toxicity, and only patients with an ECOG ≤ 1 (performance status based on the Eastern Cooperative Oncology Group) were included. 

In 2013, another step towards multiagent therapy was made when von Hoff et al. showed that gemcitabine in combination with nano-albumin-bound paclitaxel (Abraxane), compared to gemcitabine monotherapy, showed an improved overall survival (8.5 months vs. 6.7 months) for patients with metastatic PDAC [16]. 

Results of the NAPOLI 3 randomized phase III trial were recently published in LANCET. The trial evaluated the efficacy of NALIRIFOX (liposomal irinotecan, oxaliplatin, leucovorin, and fluorouracil) vs. nab-paclitaxel and gemcitabine in metastatic PDAC. A total of 770 patients with ECOG 0 or 1 were included, and no patient received prior treatment. Results showed a superior median overall survival in the patient group treated with NALIRIFOX for 11.1 months, compared to 9.2 months, in the gemcitabine plus nab-paclitaxel group [17].

Those results also led to the investigation of multiagent chemotherapies in the adjuvant setting. A comparison between modified FOLFIRINOX (=75% of the standard dose to improve tolerability and safety) and gemcitabine in the adjuvant therapy of PDAC showed a median disease-free survival of 21.6 months in the mFOLFIRINOX group, compared to 12.8 months in the gemcitabine group. Disease-free survival rate at 3 years was 39.7% for mFOLFIRINOX vs. 21.4% for gemcitabine, which led to FOLFIRINOX being the treatment of choice for ECOG 0-1 patients in the adjuvant setting [9,13,18,19]. The 5-year outcomes of this trial were published in 2021 and confirmed the improved overall survival in the mFOLFIRINOX group with 53.5 months, compared to the gemcitabine group with 35.5 months. The 5-year overall survival also showed the superiority of mFOLFIRINOX over gemcitabine (43.2% vs. 31.4%) [20].

Neoptolemos et al. investigated adjuvant therapy with gemcitabine plus capecitabine vs. gemcitabine monotherapy. The reported median overall survival for patients in the gemcitabine plus capecitabine group was 28.0 months, compared with 25.5 months in the gemcitabine group (*p* = 0.032) [21]. The strengths of this regime are better tolerability and easy administration of oral capecitabine, which makes this combination therapy the recommended treatment for patients with ECOG > 2. Altogether, multiagent chemotherapy has become the standard of care in the adjuvant and palliative settings, replacing, in most cases, monotherapy with gemcitabine [13,18]. Those encouraging results paved the way for multiple clinical trials investigating the use of new combination chemotherapies to achieve resectability in locally advanced PDAC and, more recently, in oligometastatic PDAC. 

In the following sections, we will provide a comprehensive overview of the current management of patients with locally advanced and oligometastatic pancreatic cancer.

## 2. Neoadjuvant Therapy Can Lead to Better Survival by Improving Resectability and Systemic Therapy Efficacy in Locally Advanced, but Not in Upfront Resectable Pancreatic Cancer

Neoadjuvant chemotherapy or chemoradiotherapy are already established treatment options in other entities, such as gastroesophageal cancers and rectal cancers [22,23,24].

In PDAC, the main goal is to increase eligibility for surgery for patients with initially unresectable disease by decreasing tumor size and vascular involvement and achieving higher rates of tumor-free resection margins (R0), as well as treating early systemic spread. Some have argued that neoadjuvant therapy would also facilitate systemic therapy effects, as many patients after pancreatic surgery are not able to receive adjuvant therapy [25]. The debate continues regarding whether patients with primarily resectable tumors should undergo neoadjuvant therapy. Proponents argue that this approach can reduce micrometastasis and prevent early postoperative recurrence. However, critics warn of the potential risk of disease progression during chemotherapy, which could result in missing the window for a curative resection [26].

The SWOG S1505 trial evaluated perioperative chemotherapy for resectable PDAC patients who either received mFOLFIRINOX or nab-paclitaxel followed by gemcitabine prior to surgery. A positive pathological response rate of 33% was reported. In both therapies, about 85% of patients completed chemotherapy, with around 70% of the patients proceeding to surgical resection. Median overall survival was reported to be 23.2 months for the mFOLFIRINOX group and 23.6 months for the gemcitabine/nab-paclitaxel-treated patients. The two-year overall survivals were 47% and 48% [27]. Nor-PACT, a randomized controlled study initiated by the Norwegian Gastrointestinal Cancer Group for Hepato-Pancreato-Biliary cancer, randomized patients with resectable PDAC into primary surgery or neoadjuvant therapy with FOLFIRINOX (four cycles) followed by surgery [28]. The results were presented at ASCO 2023, and median overall survival by intention to treat was 25.1 months with neoadjuvant therapy and 38.5 months in the primary surgery group (*p* = 0.096) [29]. The final publication of the results is still pending.

Altogether, available data to date do not clearly encourage neoadjuvant therapy for resectable PDAC, and international guidelines advise not to perform neoadjuvant chemotherapy in resectable PDAC [30].

## 3. Locally Advanced Pancreatic Cancer

Currently, about 30% of newly diagnosed cases of pancreatic adenocarcinoma are classified as locally advanced, and treatment recommendations vary. Multiple trials have investigated different chemotherapy regimens, as well as the duration, and appropriate measures of response aside from the radiological evaluation in neoadjuvant therapy of locally advanced PDAC.

In recent years, more patients can undergo surgical resection after neoadjuvant therapy because more complex surgical procedures, including vascular resection and replacement and multivisceral resection can be performed with adequate risk at high-volume institutions (Figure 1, Figure 2 and Figure 3). Also, in patients with arterial tumor involvement, a periarterial tumor divestment can become a feasible treatment option after neoadjuvant therapy, and arterial resection in locally advanced pancreatic cancer is an effective surgical option in specialized centers [31,32]. Also, in cases of cavernous transformations of the portal vein, different surgical approaches make a resection nowadays feasible [33].

NEOLAP, an open-label, multicenter, randomized phase 2 study from Germany evaluated different chemotherapy regimens (two cycles of nab-paclitaxel followed by randomization and either four cycles of FOLFIRINOX or two additional cycles of nab-paclitaxel) in locally advanced pancreatic cancer. The primary endpoint of the study was the surgical conversion rate. The findings suggested that nab-paclitaxel plus gemcitabine had similar effect and safety as nab-paclitaxel plus gemcitabine followed by FOLFIRINOX as a multidrug induction chemotherapy regimen for locally advanced pancreatic cancer. Patients that became eligible for surgery (about 1/3 of all patients) and subsequently underwent resection showed a significant survival benefit of 27.4 months versus 14.2 months. The overall survival for the FOLFIRINOX arm was 20.7 months, and the overall survival was 18.5 months for the Gem-NaP arm [35]. A retrospective analysis published in Annals of Surgery in 2019 analyzed 415 patients with locally advanced adenocarcinoma who underwent surgical resection after neoadjuvant chemotherapy. Patients received FOLFIRINOX, mFOLFIRINOX, or gemcitabine/nab-paclitaxel or both. Altogether, about 20% of patients became eligible for surgical resection following individual neoadjuvant therapy, and an RO resection was achieved in 89% of the cases. Patients deemed eligible for surgical resection also received radiation therapy before surgery. Median survival was significantly higher in the surgical resection group (35.3 vs. 16.2 months). The actual value of radiation therapy is hard to derive from this study, as it was added to all cases that were offered surgical exploration [36].

Hackert et al. analyzed 575 patients with locally advanced and not-resectable PDAC receiving neoadjuvant therapy (FOLFIRINOX/gemcitabine and nab-paclitaxel and others). Resection was performed in 50.8% of patients, while the resection rate following FOLFIRINOX was even higher at 60% [37].

A 2021 study investigated the effect of total neoadjuvant therapy (systemic therapy followed by chemoradiotherapy) for borderline and locally advanced adenocarcinomas. Overall, 194 of 253 patients (exclusions were made due to metastatic or unresectable disease) underwent surgical resection. A total of 63% of the included patients had borderline resectable PDAC, and 37% had locally advanced PDAC. Negative resection margins were achieved in 94%. Three factors were associated with prolonged survival: extended-duration chemotherapy (six or more cycles), good post-chemotherapy CA19-9 response, and major pathologic response. Radiologic downstaging was low, with only 28%, and thus, resectability was not reflected. The authors also encouraged explorative laparotomy in cases that do not show significant radiological downstaging but a significant drop in CA 19-9 levels [38]. This is in line with observations in other studies, where radiological response, according to RECIST, was not a good predictor for resectability, and a drop in CA-19-9 was a parameter showing the clinically relevant response much earlier [36,39].

A study from the MAYO Clinic published in 2021 investigated a therapy switch in neoadjuvant chemotherapy regimens. A total of 468 patients with borderline or locally advanced PDAC were included. A total of 70% of included patients received first-line chemotherapy (FOLFIRINOX, FOLFOX, or gemcitabine, based on the ECOG and institution), followed by surgical resection. The remaining 139 patients underwent the switch in chemotherapy to the regimen not given prior. A total of 100 patients were eligible for curative intent surgical resection following the switch in chemotherapy. The study showed no statistically significant difference in overall survival when comparing the resected groups. However, the overall survival was significantly worse for those patients who underwent a switch in chemotherapy and did not receive surgical therapy. It was suggested that the switch in chemotherapy in a neoadjuvant setting might allow a larger section of patients to reach secondary resectability [40].

All of the studies had relatively high resection rates following neoadjuvant therapy.

However, the reported resection rates showed large differences ranging from a 60% resection rate following therapy with FOLFIRINOX by Hackert et al. to 15% in an observational cohort study from Italy [37,41]. In the latter study with a low resection rate, 680 patients with borderline resectable (39.9%) and locally advanced (60.7%) PDAC were included. A total of 570 patients received chemotherapy with mostly FOLFIRINOX or gemcitabine/nab-paclitaxel. 

One reason for the different resection rates following chemotherapy (Table 2) may be a selection bias in the retrospective studies. Neoadjuvant treatment strategies also varied regarding the regimen (chemotherapy or radiochemotherapy) and duration of therapy. Furthermore, the extent of surgical resection, especially regarding vascular resection and reconstruction, varied. Downstaging criteria were also not standardized, and usually, a combination of factors was considered (radiologic response, Ca 19-9 levels), which could have excluded patients in studies where exploratory laparotomy was not routinely performed after neoadjuvant chemotherapy. 

Another limitation is what is considered resectable upon explorative laparotomy by the individual surgeon and the center’s approach. Those cases should be primarily treated in high-volume centers that have radiological and surgical expertise for locally advanced cases after neoadjuvant therapy [13]. Guidelines in Germany, as well as in other countries, now specifically address the treatment of locally advanced PDAC and recommend initial chemotherapy (FOLFIRINOX or gemcitabine plus nab-paclitaxel), as well as exploratory laparotomy, to evaluate resectability after neoadjuvant treatment [13]. The US guidelines additionally include radiotherapy [9].

The therapy of locally advanced pancreatic adenocarcinoma remains challenging, but promising steps to optimize neoadjuvant therapy and surgical outcomes have been made; more patients could be referred to surgical therapy with a subsequent improvement in overall survival. Further studies will examine new therapeutic strategies and combinations of therapies, and individualized treatment approaches for a small subgroup of patients with molecularly distinct variants (e.g., BRCA) will also play a more important role in the future.

In addition to neoadjuvant therapy, several new modalities to improve the downstaging of locally advanced adenocarcinomas are under investigation. Intraoperative applied radiofrequency ablation was evaluated for tumor downstaging but showed significant side effects [42]. However, endoscopic ultrasound-guided (EUS) radiofrequency ablation showed a high technical success rate for benign pancreatic lesions and is increasingly used in the therapy of unresectable pancreatic cancer [43]. Radiofrequency ablation could play a role in the future treatment of pancreatic adenocarcinoma [44].

Alongside therapeutic strategies, diagnostics for pancreatic adenocarcinoma are also continuously improving. New endoscopic ultrasound biopsies used to obtain tissue for primary diagnosis now allow the analysis of molecular markers, providing us with more detailed information on the underlying pathogenesis. This might help to identify prospective targets in anticancer therapy [45].

**Table 2 cancers-15-05881-t002:** Most relevant studies investigating neoadjuvant therapy in LAPC.

Reference	Type of Study	Included	Treatment Regimen	Resection Rate	Median OS (Months)
Kunzmann et al. PMID: **33338442** [35]	Open-label, randomized phase II clinical trial	LAPC	2 cycles of nab-paclitaxel–randomization–4x FOLFIRINOX or 2 cycles nab-paclitaxel	35.9%	18.5 nab-paclitaxel 20.7 FOLFIRINOX
Hewitt et al. PMID: 33630475 [46]	Phase III randomized clinical trial	LAPC + BRPC	FOLFIRINOX or gemcitabine/nab-paclitaxel followed by RCT or immunotherapy	NR	14.9 months SOC 12.4 months with immunotherapy
Murphy et al. PMID: **31145418** [47]	Single-arm phase II clinical trial	LAPC	FOLFIRINOX + losartan + RT	69% (R0)	31.4
Blazer et al. PMID: **25358667** [48]	Retrospective	LAPC + BRPC	mFOLFIRINOX + radiotherapy	51.1% ALL 44% for LAPC	21.2
Marthey et al. PMID: 25037971 [49]	Prospective, observational	LAPC	FOLFIRINOX + RCT	36.4%	22.0
Wo et al. PMID: 28134673 [50]	Retrospective	LAPC + BRPC	FOLFIRINOX + gemcitabine + RT	39.2%	18.1
Hackert et al. PMID: 27355262 [37]	Retrospective	LAPC	FORLFIRINOX or gemcitabine + RT or others	61% FOLFIRNOX 46% gemcitabine 52% others	15 with resection 8.5 only exploration
Sadot et al. PMID: 26065868 [51]	Retrospective	LAPC	FOLFIRINOX + gemcitabine + RT	30%	26 with resection 11 no resection
Gemenetzis et al. PMID: 29596120 [36]	Retrospective	LAPC	FOLFIRINOX or gemcitabine or a combination	20%	35.3 with resection 16.3 without
Maggino et al. PMID: 31339530 [41]	Prospective	LAPC + BRPC	FOLFIRINOX or gemcitabine nab-paclitaxel	15% total 9% LAPC	41.8 with resection (LAPC)

LAPC: locally advanced pancreatic adenocarcinoma; BRPC: borderline resectable pancreatic adenocarcinoma. OS: overall survival; RCT: chemoradiotherapy; RT: radiotherapy.

## 4. Oligometastatic Disease in Pancreatic Cancer

Upon primary diagnosis, 50% of patients with PDAC have metastatic disease and, therefore, are left with palliative therapy and are not eligible for surgical resection, regardless of local resectability.

The most common sites of PDAC metastasis are the liver (90%), lymph nodes (25%), lung (25%), peritoneum (20%), and bones (10–15%) [15]. The term oligometastatic disease has been coined to describe limited metastasis, mostly confined to one organ system and used in different cancer entities.

Those patients are considered to potentially benefit from an individualized therapeutic approach, including surgery, due to the limited metastatic load [52].

However, there is no universal definition for oligometastasis, and the definitions vary in literature and depending on the primary cancer site. Some studies propose less than three metastases, some less than four metastases confined to a single organ, while there are also studies including metastases to multiple sites [53,54].

For the first time, the German treatment guidelines for pancreatic cancer in 2021 defined oligometastasis as three or less synchronous metastases but recommended surgery of the primary tumor and metastases strictly in a prospective trial setting [13]. This is a paradigm shift that encourages the exploration of treatment options for these patients.

A potential benefit of the surgical therapy of oligometastatic disease has been described in other cancer entities. In a review about oligometastatic disease of gastroesophageal carcinoma, the number of metastases was defined as less than three metastases confined to a single organ system. Local therapy with surgery or stereotactic radiation of oligometastatic disease was superior regarding overall survival, compared to systemic therapy alone [55,56]. In breast cancer, oligometastasized disease (defined as no more than five metastases) was first mentioned in 2007, and a multidisciplinary strategy including local treatment of metastasis with surgery or stereotactic radiation is recommended [53,57].

Recent literature proposed defining oligometastatic disease in patients with PDAC not only by the number of metastases but by anatomical and biological criteria.

The criteria included limited disease (no more than four metastases), limited extent of necessary hepatic resection, and CA 19-9 levels below 1000 U/mL. The goal was to identify patients with a favorable tumor biology benefitting the most from the individualized treatments approach. Chemotherapy before evaluation for surgery should always be performed [12]. Several retrospective studies with a limited number of patients are available, and some show a beneficial effect of an individualized surgical approach for patients with oligometastatic pancreatic adenocarcinoma on survival [58,59,60]. In most of these studies, metastatic spread to the liver is analyzed.

In 2005, a case control study including 42 patients with oligometastatic disease was published. Oligometastasis was defined as two or less metastatic lesions smaller than 4 cm in the liver or lung. Six patients with oligometastatic disease (M1) underwent surgical resection, including metastasectomy and/or radiofrequency ablation (RFA) after neoadjuvant therapy. Overall median survival in the M1 surgery group was higher than in the M1 no-surgery group (2.7 vs. 0.98 years) and was similar to the resection group without metastasis (2.7 vs. 2.02 years) [58].

In 2016, Tachezy et al. performed a multicenter retrospective analysis including 69 patients who underwent simultaneous resection for primary tumor and liver metastasis. The number of metastases ranged from two to eleven. This was compared to a group of patients receiving exploration with palliative bypass surgery. Overall survival was significantly higher in the resected group (14.5 vs. 7.5 months). Patients included in this study received no therapy prior to the operation [59].

This was followed by a 2017 analysis by Hackert et al. of PDAC patients with limited metastasis who had undergone primary tumor and metastasis resection (liver and distant aortocaval lymph nodes). A total of 128 patients were included (intention-to-treat, oligometastatic stage; liver *n* = 85; ILN (interaortocaval lymph nodes) *n* = 43). Surgical morbidity and 30-day mortality following the synchronous resection of the primary tumor and metastasis were reported to be 45% and 2.9%, respectively. Overall median survival after resection was 12.3 months in both groups. The 5-year survival was reported to be 8.1% after surgery for liver metastasis and 10.1% following the resection of the primary tumor and ILN [60].

Yang et al. also reported a better overall survival in patients with synchronous liver metastasis resection for oligometastasized vs. non-oligometastasized patients who underwent resection (more than three liver metastases or liver metastasis with multivisceral resection) with PDAC vs. palliative systemic therapy without resection (16.8 months vs. 7.05 months vs. 8 months) [61].

Aside from the liver, metastases to the lung have been studied regarding surgical options for resection in PDAC in the oligometastatic setting.

A review analyzed metachronous resection in patients who previously underwent curative resection of PDAC. From 15 included patients, 11 showed resectable metastases. Low perioperative morbidity (8%) and no mortality were reported in this small cohort.

Median disease-free survival (DFS) and overall survival (OS) after pulmonary metastasis diagnosis were 18 months and 26 months, respectively. It was concluded that metachronous resection can be performed safely and effectively [62]. Another study retrospectively evaluated 159 patients who underwent curative intent surgery for PDAC and further analyzed patients who developed metachronous pulmonary metastasis. Out of 20 patients with pulmonary metastasis, three showed resectable disease. Two of these patients underwent surgical resection, and by the time of publication, they showed post-surgery disease-free survivals of 11 and 13 months. It was noticeable that isolated pulmonary metastases had a prior disease-free survival and overall survival of 35.4 and 81.4 months, respectively. Comparing this to patients with non-pulmonary metastasis with a prior DFS of 9.4 and a 15.8 overall survival, the prognosis in pulmonary metastasis seems much better than metastases to other sites [63].

No data exist on synchronous pulmonary resection yet. The role of surgery in pulmonary metastases is not clear because patients with only pulmonary metastases show improved survival, and current research assumes that pulmonary metastasis-only PDAC has a different tumor biology.

In a comprehensive literature search analyzing PubMed and Cochrane databases, 428 patients who underwent surgical resection for liver metastasis s (*n* = 343), lung metastasis (*n* = 57), and peritoneal dissemination (*n* = 28) were analyzed. The studies analyzed for the synchronous resection of liver metastasis following the response to initial chemotherapy reported median overall survivals of 27 and 34 months. For metachronous lung metastasis, the median overall survival ranged from 51 to 121 months [64].

All of the mentioned studies analyzing synchronous hepatic metastasis resection concluded a potential overall survival benefit in synchronous resection. However, all of them were retrospective analyses, with a potentially high selection bias and a relatively small number of patients.

The HOLIPANC (hepatic oligometastatic adenocarcinoma of the pancreas) trial is the first prospective phase II trial for patients with oligometastasized PDAC initiated in Germany and is currently recruiting. Patients with a maximum of five liver metastases are eligible for inclusion. Patients receive a combination of liposomal irinotecan (nal-IRI), oxaliplatin (OX), and 5-fluouracil (5-FU)/folinic acid (FA) (nal-IRI + OX + 5-FU/FA, NAPOX) as neoadjuvant chemotherapy and are referred to exploratory laparotomy with the goal of primary tumor and metastasis resections when subsequent staging shows a response or stable disease [10]. Currently, the randomized controlled phase III METAPANC trial has been funded in Germany, which will compare resection vs. non-resection after a FOLFIRINOX induction chemotherapy in oligometastatic PDAC patients [65].

In 2018, a randomized controlled trial was initiated in China evaluating the synchronous resection of oligometastatic (no more than three metastases to the liver) PDAC following induction chemotherapy. The study is planned to finish recruiting in 2023 [11].

The ScanPan 1 trial, a Scandinavian prospective multicenter study, plans to assess the curative-intent multimodal treatment protocol for patients with oligometastatic pancreatic cancer. The study will investigate two cohorts. The first cohort is comprised of patients with hepatic metastases from PDAC without metastases to other organs subclassified into limited (less than four metastases smaller than 5 cm) and extensive diseases. The other cohort will be patients with unilocular metastases to any intra- or extra-abdominal location that could potentially be treated with stereotactic radiation, surgery, thermal ablation, or a combination of therapies [66]. The fact that several prospective trials are initiated globally once more depicts the importance of the subject. Those trials will be able to help answer the most pressing questions when considering surgical therapy in patients with metastasized PDAC, such as the best choice of neoadjuvant treatment, number of metastases considered oligometastatic, other clinical inclusion criteria, and extent of surgical resection (Table 3).

## 5. Conclusions and Outlook

Considering stage IV PDAC patients for surgical resection represents a paradigm shift. Based on current literature, no recommendation can be made outside of clinical trials. But the sum of studies that have analyzed surgical resection in selected patients as well as the increasing efficacy of multiagent chemotherapy should encourage the establishment of criteria for individual patient selection to undergo aggressive surgical therapy.

This leaves us with high expectations for the results of the first randomized controlled trials.

## Figures and Tables

**Figure 1 cancers-15-05881-f001:**
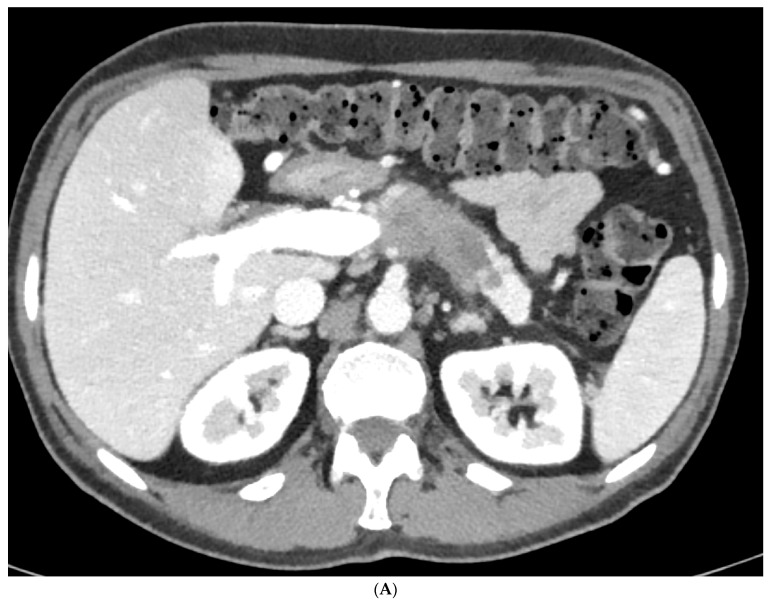
(**A**) Patient with locally advanced pancreatic body adenocarcinoma with close contact to the splenic vein and the gastroduodenal artery (arrows in (**B**) and (**C**); one arrow: gastroduodenal artery; two arrows: splenic artery). The patient received neoadjuvant-intended chemotherapy with gemcitabine + nab-paclitaxel followed by chemoradiation at the University Hospital Cologne. This was followed by surgical resection (total pancreatectomy); the histopathology showed an R0 resection.

**Figure 2 cancers-15-05881-f002:**
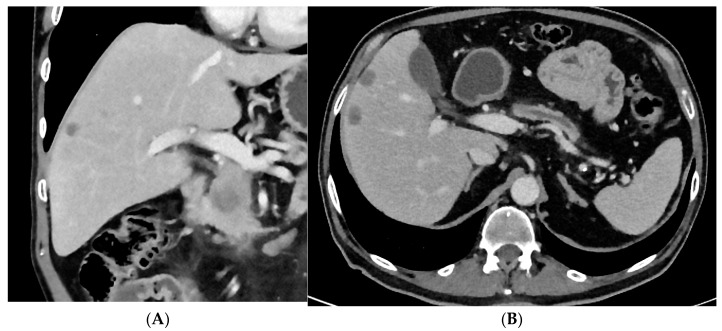
A patient with three liver metastases with adenocarcinoma of the pancreatic head. (**A**) Coronal view with pancreatic head carcinoma and liver metastasis. (**B**) Axial view showing liver metastasis and dilation of the pancreatic duct.

**Figure 3 cancers-15-05881-f003:**
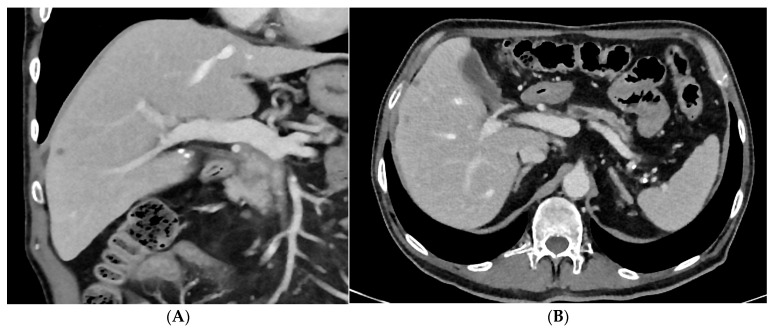
The same patient with therapeutic response after therapy (liposomal irinotecan +folinic acid + 5 FU NAPOLI) treated within the FOOTPATH study [34]. Due to good radiologic response, the individual therapeutic concept of surgical resection was recommended by a multidisciplinary tumor conference. The patient also underwent surgical resection at the University Hospital Cologne (pancreatic head resection with atypical hepatic resection of liver segments V and VI and microwave ablation of one metastasis in segment VI). Histology showed ypT1c, ypN0, ypM1, L0, V0, Pn1, and R0. (**A**) Coronal view post chemotherapy showing a decrease in size of the liver metastasis seen previously (Figure 2A), (**B**) Axial view after chemotherapy showing therapeutic response of the liver metastases.

**Table 1 cancers-15-05881-t001:** Resectability criteria modified according to NCCN guidelines [9].

	Arterial Involvement			Venous Involvement
Resectability	Common Hepatic Artery	Superior Mesenteric Artery	Celiac Artery	Portal/Superior Mesenteric Vein
Resectable	No tumor contact	No tumor contact	No tumor contact	No tumor contact or ≤180° contact without irregularity of the vein
Borderline resectable	Solid tumor contact without the involvement of the celiac trunk or hepatic artery bifurcation	≤180°	≤180° ≥180° without the involvement of the aorta or gastroduodenal A involvement (body/tail)	≥180° or ≤180° with contour irregularity or thrombosis with reconstructable PV/SMV Solid tumor contact with IVC
Locally advanced		>180°	>180° (head/uncinate) Solid tumor contacts with CA and aorta	Unreconstructable portal vein or superior mesenteric vein due to tumor involvement or thrombosis/occlusion

**Table 3 cancers-15-05881-t003:** Prospective, multicenter trials investigating therapy of oligometastasized pancreatic adenocarcinoma.

TRIAL	REFERENCE	DESIGN	TYPE OF STUDY	INCLUSION CRITERIA	PRIMARY ENDPOINT/Objective	ARMS	KEY FINDINGS
HOLIPANC	PMID: 34794396	PROSPECTIVE, MULTICENTER, NON- RANDOMIZED	PHASE II	OLIGOMET. PDAC (1–5 HEPATIC MET.)	OS AFTER R0/R1 RESECTION	SINGLE ARM	ONGOING
CSPAC-1	PMID: 31818843	PROSPECTIVE, MULTICENTER, RANDOMIZED	PHASE III	OLIGOMET. PDAC (≤3 HEPATIC MET.)	ROS IN RESECTED GROUP AFTER CHEMOTHERAPY VS. CHEMOTHERAPY GROUP	DOUBLE ARM	ONGOING
SCANPAN 1	NCT05271110	PROSPECTIVE, MULTICENTER	PROSPECTIVE COHORT STUDY WITH A SINGLE GROUP ASIGNMENT	COHORT 1: SYN- OR METACHAROUNOUS HEP. METASTASIS COHORT 2: SYN- OR METACHRONOUS HEPATIC MET. + AT LEAST ONE EXTRHEPATIC MANIFESTATION	SAFETY, FEASIBILITY, TOLERABILITY, AND CLINICAL OUTCOMES	SINGLE ARM	ONGOING

PDAC: Pancreatic ductal adenocarcinoma; OS: overall survival; ROS: real overall survival (time from diagnosis to death).

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
