# Peer review of "Therapy of Locally Advanced and Oligometastatic Pancreatic Adenocarcinoma"

_cancers, 2023, doi:10.3390/cancers15245881_

Round 1

Reviewer 1 Report

Comments and Suggestions for Authors

Overall, I congratulate the authors on a thorough review of the literature surrounding the management of locally advanced and metastatic pancreatic cancer. I believe this is a comprehensive piece of work, that warrants publication, but this requires some improvements prior to being publication ready. This includes a number of spelling errors. I have summarised these in the following section. Further suggestions can be found below:

1. Table 1 has been poorly edited - I am not sure if this is due to the journal editing process, but it makes it challenging for a reader to assess what criteria is in which box.

2. I think the readers would really benefit from figures demonstrating the difference between resectable, borderline resectable and locally advanced PC. As surgeons we understand and appreciate the intricacies of these criteria, and why they are important, but the average reader will not.

3. A table summarising any trials in locally advanced PC would be great. What was the design, outcome measure, and outcome? Just to summarise the data for the reader.

4. Similarly, a table summarising the prospective trials for oligo-metastatic PC would be extremely beneficial. I recommend including design, outcome measures, etc.

Comments on the Quality of English Language

There are numerous spelling errors throughout the manuscript. I recommend the authors carefully review the spelling and grammar and I have summarised the errors I found below:

Page 1, para 3: "Goldstandard" should read "The gold standard"

Page 1, last line: "AND" should not be used to start a sentence

Pg2 para2: "Main driver" should read "The main driver"

Pg 4 Para2: "adeuate" should read "adequate"

Pg5: The figure legend needs to be improved and some of the legend has been blended into the main text

Pg6 Para 2: "Chemotherapie"

Pg6 para2: The sentence "The remaining 139..." does not read right and I suggest re-writing

Pg6 Para3: "larg" and para4: "resetable"

Pg8 Para 2: The sentence "Those included limited.." needs re-written

Author Response

Response to Reviewer I Comments

Point-by-point response to Comments and Suggestions for Authors

Comments 1: 1. Table 1 has been poorly edited - I am not sure if this is due to the journal editing process, but it makes it challenging for a reader to assess what criteria is in which box

Response 1: Thank you for pointing this out. We agree with this comment.

It seems that due to the upload the divisions and color highlighting were lost. The table was edited.

Comments 2: I think the readers would really benefit from figures demonstrating the difference between resectable, borderline resectable and locally advanced PC. As surgeons we understand and appreciate the intricacies of these criteria, and why they are important, but the average reader will not.

Response 2: We find this difficult to graphically demonstrate as each patient has an individual anatomy. We tried to show the resectability criteria in Table 1 and with the CT-Scans.

Comments 3:

3. A table summarising any trials in locally advanced PC would be great. What was the design, outcome measure, and outcome? Just to summarise the data for the reader.

Response 3: We added a table summarizing the most relevant LAPDAC trials.

Comments 4:

4. Similarly, a table summarising the prospective trials for oligo-metastatic PC would be extremely beneficial. I recommend including design, outcome measures, etc.

Response 4: We added a table summarizing the most relevant LAPDAC trials.

4. Response to Comments on the Quality of English Language

Quality of language

Point 1: There are numerous spelling errors throughout the manuscript. I recommend the authors carefully review the spelling and grammar and I have summarised the errors I found below:

Response 1:  All listed errors were edited.

5. Further clarifications:

Another in our opinion important trial was added – Scan Pan (Page 12, highlighted).

Reviewer 2 Report

Comments and Suggestions for Authors

Very interesting and well written review.

The authors should add some comments on the potential (palliative) role of EUS-guided ablative treatments in these patients, for example radiofrequency ablation.

I recommend to comment the importance of the analysis of molecular markers that can be dosed in the sampled tissues obtained with newer EUS-guided end-cutting fine-needle biopsy needles (cite PMID: 31031330)

Author Response

Response to Reviewer II Comments

Point-by-point response to Comments and suggestions for Authors.

Comments 1: The authors should add some comments on the potential (palliative) role of EUS-guided ablative treatments in these patients, for example radiofrequency ablation.

Response 1: The topic was added. Please find it in the highlighted section on page 7&8.

Comments 2: I recommend to comment the importance of the analysis of molecular markers that can be dosed in the sampled tissues obtained with newer EUS-guided end-cutting fine-needle biopsy needles (cite PMID: 31031330)

Response 2:  The recommended study was also included. Please find it in the highlighted section on page 7&8.

5. Further clarifications:

Another in our opinion important trial was added – Scan Pan (Page 12, highlighted).

Reviewer 3 Report

Comments and Suggestions for Authors

 Therapy of Locally Advanced and Oligometastatic Pancreatic Adenocarcinoma, reviews treatment of more advanced pancreatic cancers, evaluating the role of surgical reaction following neoadjuvant therapies, where the prior treatments may have positively altered the possibility of resection.

While the message comes through and generally supports the overall claim that surgical reaction should be re-examined after prior therapies, problems with the writing and presentation  make it difficult read the review for many details presented in support of the underlying message.

Table 1 is poorly designed.  It needs some gaps  to prevent the information in columns 4 and 5 from bleeding together.

There are numerous spurious paragraph breaks.  Just after Table 1—LAN-CET.

                The trial evaluated---

There are many other such cases.

The title of section 2 seems a little misleading given the final sentence of the section.

In this section there seems to be another bad paragraph break---treated patients.

                The two year overall---, which runs into another section that might warrant a paragraph break—Nor-PACT, a ----

At the end of this section, the references go from being before the periods at the end of sentences to after.

Beginning at this point there are several misspellings of words, adeaute,  margings, morbitdity, larg, extend (t), commen.

In Figure 1, please indicate to what the arrows are pointing.

It appears if some text might have been lost when inserting Figure 1 into the paper.  We have the seemingly hanging sentence—This was followed by surgical---.

I am not sure what is meant by Low perioperative morbid(t)dity (8%) and no morbidity was reported (bottom of page 8.

Comments on the Quality of English Language

poor

Author Response

Response to Reviewer III Comments

Point-by-point response to Comments and Suggestions for Authors

Comments 1: Table 1 is poorly designed.  It needs some gaps to prevent the information in columns 4 and 5 from bleeding together.

Response 1: Thank you for pointing this out. We agree with this comment.

It seems that due to the editing process the divisions and color highlighting were lost. The table was edited.

Comments 2: There are numerous spurious paragraph breaks.  Just after Table 1—LAN-CET.

                The trial evaluated---

There are many other such cases.

Response 2:  We also agree with this comment. Paragraph breaks were edited.

Comments 3:

The title of section 2 seems a little misleading given the final sentence of the section.

Response 3:  The title of section 2 was edited.

Comments 4:

In this section there seems to be another bad paragraph break---treated patients.

                The two year overall---, which runs into another section that might warrant a paragraph break—Nor-PACT, a ----

Response 4:  We agree. Corrections were made.

Comments 5: At the end of this section, the references go from being before the periods at the end of sentences to after.

Beginning at this point there are several misspellings of words, adeaute,  margings, morbitdity, larg, extend (t), commen.

Response 5:  We agree. Corrections were made.

Response to Comments on the Quality of English Language

Point 1: poor.

Response 1:  It seems that due to the uploading/editing process numerous paragraph breaks appeared. All listed errors were edited.

Round 2

Reviewer 2 Report

Comments and Suggestions for Authors

The revised version of the paper is OK. Thank you!

Reviewer 3 Report

Comments and Suggestions for Authors

My comments were addressed.

Comments on the Quality of English Language

ok